

# Characteristics and factors influencing the natural regeneration of *Larix principis-rupprechtii* seedlings in northern China

Weiwen Zhao, Wenjun Liang, Youzhi Han and Xi Wei

College of Forestry, Shanxi Agricultural University, Taigu, China

## ABSTRACT

*Larix principis-rupprechtii* is an important and widely distributed species in the mountains of northern China. However, it has inefficient natural regeneration in many stands and difficulty recruiting seedlings and saplings. In this study, we selected six plots with improved naturally-regenerated *L. principis-rupprechtii* seedlings. A point pattern analysis (pair-correlation function) was applied to identify the spatial distribution pattern and correlation between adult trees and regenerated seedlings mapped through X/Y coordinates. Several possible influencing factors of *L. principis-rupprechtii* seedlings' natural regeneration were also investigated. The results showed that the spatial distribution patterns of *Larix principis-rupprechtii* seedlings were concentrated 0–5 m around adult trees when considering the main univariate distribution type of regeneration. There was a positive correlation at a scale of 1.5–4 m between seedlings and adult trees according to bivariate analyses. When the scale was increased, these relationships were no longer significant. Generally, adult trees raised regenerated *L. principis-rupprechtii* seedlings at a scale of 1.5–4 m. Principal component analysis showed that the understory herb diversity and litter layer had a negative correlation with the number of regenerated seedlings. There was also a weak relationship between regenerated numbers and canopy density. This study demonstrated that the main factors promoting natural regeneration were litter thickness, herb diversity, and the distance between adult trees and regenerated seedlings. Additionally, these findings will provide a basis for the late-stage and practical management of natural regeneration in northern China's mountain ranges.

## INTRODUCTION

Plants are vital for the soil-plant-atmosphere–continuum (SPAC), which is responsible for the production of high quantities of timber and other practical products (*Liu, Yu & Jia, 2019*; *Peng et al., 2008*; *Wei et al., 2010*). Previous research showed that forest reconstruction depends on natural recruitment across all plantations (*Dong et al., 2018*; *Man, Rice & Blake, 2013*; *Mc et al., 2015*; *Seiwa et al., 2012*; *Stoyan, Stoyan & Helga, 1994*), and also suggested that natural regeneration was very effective in forest development. Regeneration and

Corresponding authors
Wenjun Liang,
liangwenjun123@163.com
Youzhi Han, hanyouzhi@sxau.edu.cn

development restore degraded forest ecosystems to ones with high biodiversity and an ecological quality that is in line with those undergoing natural regeneration, which can save time and costs and increase biodiversity (*Ram et al., 2014*). Seedlings are more stable in forest ecosystems following natural regeneration. However, the regeneration in most plantations has become difficult due to artificial intervention and over-exploitation (*Gao, Li & Zhao, 2020*; *Hernandez-Barrios, Anten & Martinez-Ramos, 2015*; *Wang et al., 2017*), which cause spatial condition stress for regenerated seedlings and intense thinning density, which removes the available seed source (*Guo & Wang, 2019*). Previous studies have shown that natural regeneration is more efficient than traditional planting at maintaining ecological sustainability because of the increase of young trees and the replacement of old and dead trees (*Singh, Malik & Sharma, 2016*; *Christian et al., 2018*; *Singh & Rawat, 2012*). Therefore, it is necessary to investigate naturally regenerated seedlings when aiming to balance production, recover ecosystems, and provide solutions for forest management.

To identify the factors that affect natural regeneration, many researchers have studied stand structure, canopy density, canopy gap, and thinning intensity in forests and found that the decrease of regenerated seedlings were affected by increases in canopy density, canopy gaps, and thinning intensity (*Chen & Cao, 2014*; *Wang et al., 2017*; *Wang et al., 2019*). *Larix principis-rupprechtii* is an important species when discussing afforestation due to its high growth rate, good soil erosion resistance, fine drought tolerance, and wide distribution in the mountains of northern China. The regeneration failures of *L. principis-rupprechtii* plantations from the germination to seedling maturity stages have become a crucial issue in China (*Wenjun & Xi, 2020*), but the characteristics and influencing factors of their seedlings' natural regeneration have remained undetermined. Therefore, this study was conducted to analyze the spatial patterns between *L. principis-rupprechtii* seedlings and adult trees, as well as the additional factors that affect its regeneration.

The spatial pattern of a species is very important for its ecological processes. Studies have suggested the importance of regeneration spatial patterns for plant dynamics (*Camarero et al., 2010*; *Crawford & Hoagland, 2010*). Analyzing the spatial patterns among different species can explain the influence of adult trees on seedlings. There are also many biotic and abiotic factors affecting tree species distribution and affluence (*Cheng et al., 2014*; *Chu, Wang & Zhang, 2014*), such as litter thickness and herb coverage, which are controlled by the mechanisms of seed diffusion and the appropriate microsites for regeneration (*Devaney, Jansen & Whelan, 2014*). There are two forms of population ecological relationships in spatial patterns: spatial distribution and spatial correlation. Studies have shown that aggregated, random, and regular spatial distributions are positively, negatively, and insignificantly correlated, respectively, with spatial relationships, which explains the spatially horizontal structure and interactions between population and environmental factors (*Seidler & Plotkin, 2006*; *Wang et al., 2010*). Although the number of scattered seeds from most plant species tends to decrease with the increased distance from adult trees, the negative relationship between the number of seedlings and the distance from adult trees has not been confirmed (*Devaney, Jansen & Whelan, 2014*; *Janzen, 1970*). Spatial patterns can reveal the potential factors of these distribution patterns and their correlation, as

well as help predict the direction of population evolution processes, such as recruitment, regeneration, and dying.

Traditional analysis methods based on spatial distribution patterns, such as the non-sample method and variance block analysis, are not able to fully reflect spatial distribution because spatial patterns and correlations depend on population density and spatial scale (*Haase, 1995*; *He, Legendre & Lafrankie, 1997*; *Itoh et al., 1997*). To eliminate the limitations of these methods, *Ferrier et al. (2002)* and *Ripley (1977)* proposed an improved method based on individual digital plant location information, where they took each individual as a point in two-dimensional space, printed the spatial distribution map, and analyzed the spatial pattern. This method has overcome the shortcomings of the single scale used by traditional methods, and it also can provide comprehensive information on the distance between points and spatial pattern characteristics at any scale (*Frost & Rydin, 2000*; *Zhang & Meng, 2007*). Point pattern analysis provides a beneficial construction to analyze discrete points in space (*Wiegand et al., 2006*). Some researchers have extended it to include the K function K(r) and pair-correlation function g(r), which have been confirmed as valuable in forest ecology (*Ripley, 1977*; *Hambly, Stoyan & Stoyan, 1994*; *Stoyan, Stoyan & Helga, 1994*). *Barbeito et al. (2008)* used spatial point analysis to study canopies in different aged structures and found that an uneven and multi-aged management system was the best for regenerated seedlings. Therefore, in this study, we used a spatial point pattern to analyze the relationship between seedlings and adult trees and the distance between them.

In this study, the pair-correlation function of the spatial point pattern analysis method was adopted to analyze the spatial distribution pattern and correlation between *L. principis-rupprechtii* adult trees and seedlings. Principal component analysis was used to identify the factors that affect *L. principis-rupprechtii* regeneration. Additionally, three hypotheses were proposed: (1) adult trees can promote seedling regeneration within a certain distance; (2) understory vegetation affects the regeneration of seedlings; and (3) other environmental factors affect the natural regeneration of seedlings. In the study area, regenerated *L. principis-rupprechtii* occupied a small region with only a few adult trees. To test these hypotheses, we selected six sampling plots with greater number of naturally regenerated *L. principis-rupprechtii* seedlings in the Guandi Mountains. After analysis, the main factors affecting seedling regeneration were identified. The findings can provide guidance for the later stage management of natural regeneration and promote the maintenance and development of *L. principis-rupprechtii* plantations in the mountains of northern China.

## MATERIALS & METHODS
### Study site
This study was conducted in the Guandi Mountains, Jiaocheng county, Lvliang country, Shanxi Province (111°22′–111°33′E, 37°45′–37°55′N) (*Yang et al., 2017*). The annual, minimum, and maximum temperatures are 4.3 °C, –29.1 °C, and 17.7 °C, respectively. The average, minimum, and maximum atmospheric pressure is 936.6 hPa, 936.4 hPa, and 936.8 hPa, respectively. The average water vapor pressure is 5.3 hPa. The average annual precipitation is 822 mm, and evaporation is 1,268 mm. The altitude is between 1,500

and 2,831 m. The area has a temperate continental monsoon climate. The soil is mainly mountain cinnamon soil, cinnamon soil, and brown soil (*Yang et al., 2017*).

The forests are made up of dominant plants (*L. principis-rupprechtii*, *Pinus tabuliformis*, *Picea wilsonii*, *Picea meyeri*, *Picea asperata,* and *Populus simonii*), shrubs (*Acer tataricum subsp. ginnala*, *Rosa multiflora*, *Spiraea salicifolia*, Clematis florida, and *Berberis amurensis*), and grass (*Fragaria orientalis*, *Geranium wilfordii*, *Chrysanthemum chanetii*, *Lathyrus humilis*, *Rubia cordifolia*, *Bupleurum smithii*, and *Thalictrum aquilegiifolium*).

## Experimental design

We investigated *L. principis-rupprechtii* seedling regeneration in the Guandi Mountains in a research area of 2 hm$^2$ in August 2020. The forest area was a plantation that did not have artificial intervention or management. Six fixed 40×40 m plots were selected (Fig. 1). The regeneration degree was evaluated (high = number of regenerated seedlings >5,000 tree/ha, medium = number of regenerated seedlings between 3,000 and 5,000 tree/ha, and low = number of regenerated seedlings between 2,500 and 3,000 tree/ha). In order to efficiently identify useful methods to stimulate natural regeneration, we studied the sampling plots with high regeneration densities, and then analyzed the effects of the naturally regenerated seedlings.

## Data collection

Field measurements were carried out in six plots between August 1 and August 30, 2020. After the plots were laid out, we measured their land parameters, including aspect, slope, elevation, longitude, latitude, and canopy density (Table 1). Simultaneously, all *L. principis-rupprechtii* individuals were mapped in a plane-coordinate system. The coordinates of each *L. principis-rupprechtii* individual was measured using a tape and compass. We recorded each adult tree's growth parameters, including height, diameter, breast height, and crown. The distances between *L. principis-rupprechtii* seedlings and the adult trees were recorded. Two categories were established: seedlings (0.3 m <height <2.5 m) and saplings (height ≥ 2.5 m). Five sampling points (1×1 m) in each plot were selected for herb species investigation using a X shape sampling method. Each herb species' height, number, and coverage was recorded before bagging and weighing. The significant value was adopted to measure diversity indices such as the Simpson index, Shannon–Wiener index, and Pielou index. The shrub species across five quadrats (5×5 m) were then investigated and recorded using X-shape sampling. Five 30×30 cm litter sampling points near each herb and shrub sampling plot were additionally investigated. After the stones were removed, the undecomposed and semi-decomposed litter was classified, bagged, weighed, and labeled for further laboratory analysis. Five litter samples and five herb samples were collected from each plot.

## Spatial pattern statistical analysis

Our point pattern analysis illustrated the spatial relationship and described the effect of distance between the *L. principis-rupprechtii* adult trees and seedlings. The null model identified potential differences in this spatial distribution. Pair-correlation function g(r) combined with Ripley's K function showed the spatial relationship of different species.

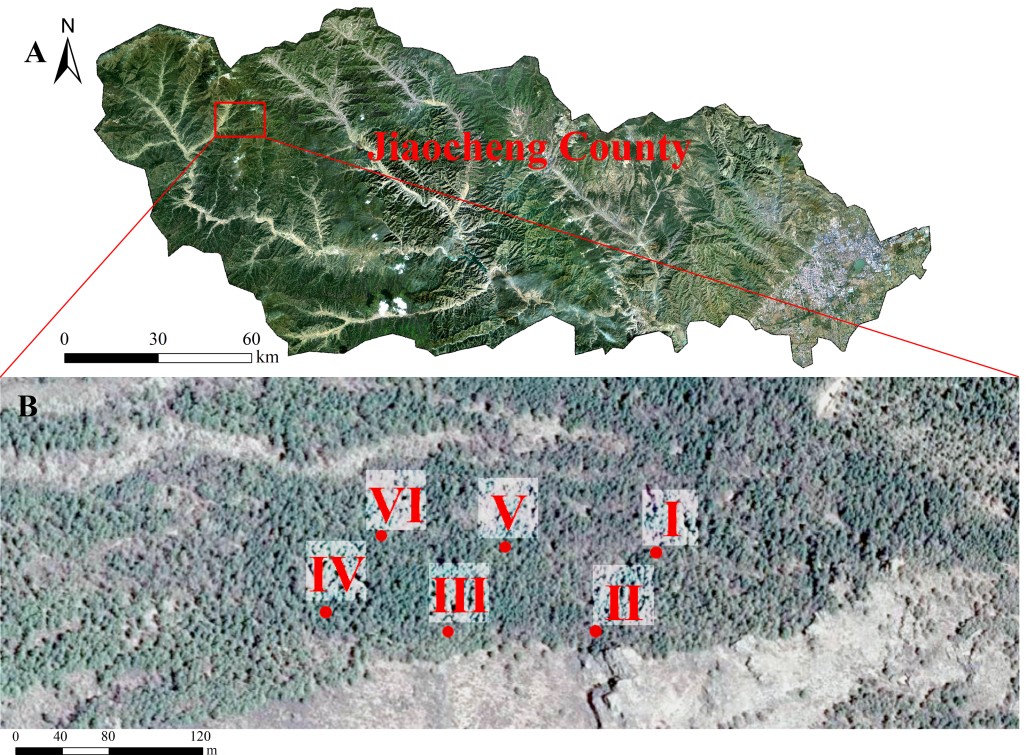

**Figure 1** **The study site, *Larix principis-rupprechtii* forests in Jiaocheng County (A). Red dots indicate the sampling plots (I-VI), gray boxes indicate the six sampling areas (B).**

The random labeling null model was utilized to complete spatial randomness (CSR). The expected number of points at distance d and a short analysis area, K(d), were calculated by:

$$\mu K(d) = \sum \sum \frac{w_{ij}(d)}{n} \tag{1}$$

where $\mu$ is the density of regenerated seedlings per plot, n is the number of trees, and $w_{ij}(d)$ is a weight function that considers the edge effect and represents the reciprocal of the number of circles centered on i (the radius is the distance from i to j) falling into the area (*Barbeito et al., 2008*).

The pair-correlation function g(r) assessed the critical range of regenerated clusters, which was more efficient compared to the K function that removed the scale dependence of K(d). The pair-correlation was the derivate of the K function, calculated by:

$$g(r) = K'(r)/2\pi r \tag{2}$$

where r is the spatial scale, and dot is the sampling circle with the target trees. g(r)>1, g(r) =1, and g(r)<1 indicated the pattern aggregation, randomness, and regularity, respectively, on the scale r.

**Table 1  Basic status of sample trees.**

| Sample plot | I | II | III | IV | V | VI |
|---|---|---|---|---|---|---|
| Tree age (a) | 59 | 50 | 62 | 64 | 65 | 59 |
| Longitude | 111°32′33″ | 111°32′28″ | 111°32′25″ | 111°32′24″ | 111°32′26″ | 111°32′25″ |
| Latitude | 37°51′21″ | 37°51′19″ | 37°51′17″ | 37°51′15″ | 37°51′20″ | 37°51′20″ |
| Altitude (m) | 2066 | 2060 | 2029 | 2018 | 2025 | 2023 |
| Slope (°) | 13 | 23 | 25 | 22 | 18 | 21 |
| Aspect | NW | NW | NW | NW | NW | NW |
| Density (tree/ha) | 125 | 250 | 175 | 125 | 50 | 50 |
| Crown density (%) | 30 | 20 | 10 | 10 | 50 | 20 |
| DBH (cm) | 35.7 ± 2.9 | 31.5 ± 8.0 | 24.1 ± 13.6 | 32.5 ± 12.2 | 34.7 ± 2.1 | 37.9 ± 2.2 |
| Height (m) | 23.9 ± 3.0 | 17.8 ± 5.4 | 19.8 ± 8.7 | 16.5 ± 3.7 | 14.5 ± 0.7 | 24.0 ± 0.1 |
| Regeneration density (tree/ha) | 6000 | 8175 | 10725 | 11225 | 9325 | 5500 |
| Main Vegetation | *Rosa multiflora, Acer tataricum, Geranium wilfordii, Chrysanthemum chanetii, Thymus mongolocus* | *Acer tataricum, Berberis amurensis, Aster trinervius, Chrysanthemum chanetii* | *Spiraea salicifolia, Acer tataricum, Rosa multiflora, Fragaria orientalis, Lathyrus humilis* | *Rosa multiflora, Acer tataricum, Chrysanthemum chanetii, Fragaria orientalis, Carex duriuscula* | *Spiraea salicifolia, Acer tataricum, Rosa multiflora, Berberis amurensis, Astragalus mongholicus* | *Rosa multiflora, Spiraea salicifolia, Acer tataricum, Vicia amoena, Fragaria orientalis* |

**Notes.**

NW is the northwest aspect of each plot.

The spatial relationship between seedlings and trees was analyzed using the $g_{12}(r)$ function:

$$g_{12}(r) = dK_{12}(r)/(2\pi r \cdot d(r)) \tag{3}$$

where $g_{12}(r) = 1$, $g_{12}(r) > 1$, and $g_{12}(r) < 1$ indicate insignificant, positive, and negative correlations, respectively, between two different species on the scale r. These species were selected and combined within the two defined categories (seedlings and adult trees) to analyze their relationship.

The spatial relationship between the regenerated seedlings and adult trees was examined using CSR. Random labeling is not commonly used in sample plots. Consequently, we adopted a case-control design with *L. principis-rupprechtii* seedlings as pattern 1 (the control pattern) and *L. principis-rupprechtii* adult trees as pattern 2 (case). $g_{12}(r)$ provided insight into the spatial distribution of *L. principis-rupprechtii* seedlings around adult trees. At a given scale, if $g_{12}(r)$ was above the confidence envelope, *L. principis-rupprechtii* seedlings were positively associated with *L. principis-rupprechtii* adult trees on the scale r. However, if $g_{12}(r)$ was below the confidence envelope, that showed a negative association between seedlings and trees.

All analyses were conducted using the Programita software package and Origin 2021 software (*Wiegand & Moloney, 2004*). The scale r represented spatial scales within 20 m (calculated up to half of the shortest side of the sampling plots). We used a 1 m cell scale

and calculated the statistics up to a scale r of 20 m in view of plot size. One hundred and ninety-nine Monte Carlo simulations with a significance level of 99% were used to determine whether the species distribution pattern and correlation were significant (*Wiegand & Moloney, 2004*).

## RESULTS

### Spatial distribution patterns of *L. principis-rupprechtii* seedling populations across different sampling plots

Only seedlings of the same age and adult trees were found in the sampling plots, and no saplings were found. We observed that *L. principis-rupprechtii* seedlings and *L. principis-rupprechtii* adult trees were randomly distributed (I–VI) (Fig. 2). There was low spatial heterogeneity between seedlings and trees. We additionally explored *L. principis-rupprechtii* seedling spatial patterns across six sampling plots, and the g(r) function analysis results showed that the spatial distribution patterns changed with the spatial scale (Fig. 3). *L. principis-rupprechtii* seedlings in plot I showed aggregated distribution at a scale of 0–17.5 m, and random distribution at a scale of 17.5–20 m (Fig. 3–I). *L. principis-rupprechtii* seedlings in plot II showed aggregated distribution at a scale of 0–15 m (Fig. 3–II). *L. principis-rupprechtii* seedlings in plot III showed aggregated distribution at a scale of 0–9 m, random distribution at a scale of 10–12.5 m, and regular distribution at a scale of 12.5–20 m (Fig. 3–III). *L. principis-rupprechtii* seedlings in plot IV showed aggregated distribution at scales of 0–5 m and 7.8–17.5 m, and random distribution at a scale of 5–7.8 m (Fig. 3–IV). *L. principis-rupprechtii* seedlings in plot V showed aggregated distribution at scales of 0–9.5 m and 10–19 m (Fig. 3–V). *L. principis-rupprechtii* seedlings in plot VI showed aggregated distribution at a scale of 0–5 m and random distribution at a scale of 5–20 m (Fig. 3–VI). Generally, *L. principis-rupprechtii* seedlings showed aggregated distribution at small scales and random or regular distribution with increased spatial scale. The scatter distribution of populations in the sampling plots (Fig. 3) also showed that *L. principis-rupprechtii* seedlings in different sample plots were clustered in the Guandi Mountains.

### Bivariate spatial analysis of *L. principis-rupprechtii* seedlings and adult trees in different sampling plots

Our spatial correlation analysis of *L. principis-rupprechtii* adult trees and seedlings showed that their spatial correlations (Fig. 4) were different across these six sampling plots in the Guandi Mountains. In sample plot I, *L. principis-rupprechtii* seedlings and *L. principis-rupprechtii* adult trees were positively correlated at a scale of 0–1 m, and showed weakly positive correlation at scales of 3.5–6.5 m and 10–15 m. Furthermore, there was no correlation at the other scales (Fig. 4–I). In sample plot II, there was a slightly negative correlation at a scale greater than 19 m. *L. principis-rupprechtii* seedlings and adult trees showed a negative correlation at a scale of 17.5–18.5 m and no correlation in any other scales (Fig. 4–II). In sample plot III, there was a negative correlation at scales of 0–1.5 m, 5–6.5 m, and 13–15.2 m, and no correlation in other scales (Fig. 4–III). In sample plot IV, there was a positive correlation at scales of 1.5–2.8 m and 8.2–10 m, and there was no correlation at scales of 0–1.5 m, 2.5–8.2 m, and 10–20 m between *L. principis-rupprechtii*

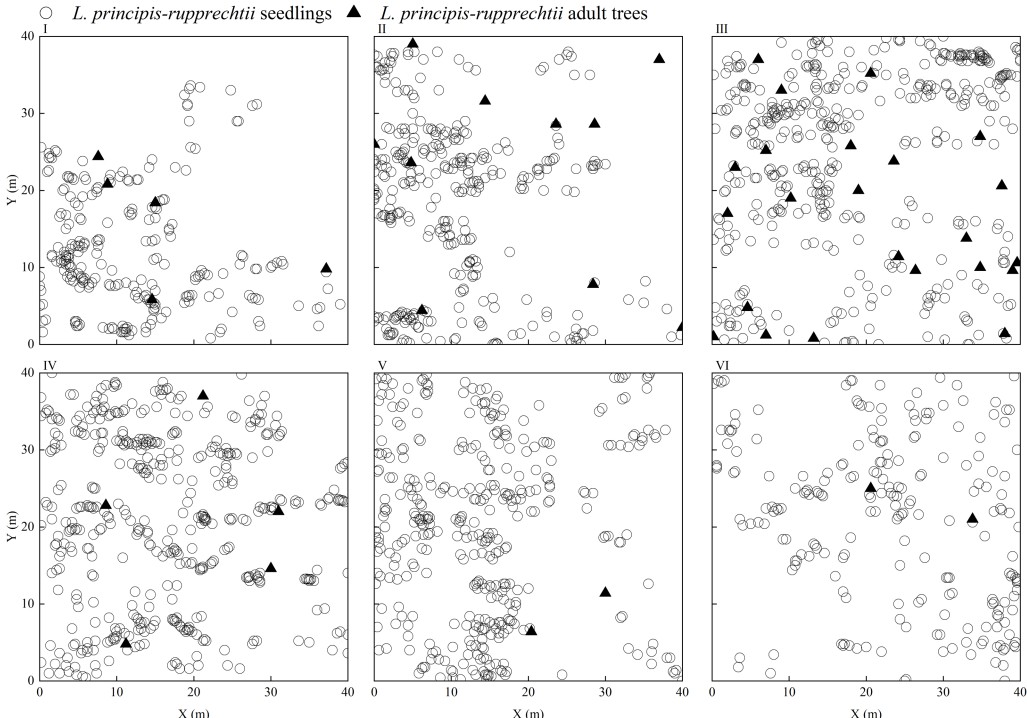

**Figure 2** **The distribution of *L. principis-rupprechtii* seedlings and *L. principis-rupprechtii* adult trees in the six sampling plots (I, II, III, IV, V, and VI).** *L. principis-rupprechtii* seedlings are represented by a circle (○) and *L. principis-rupprechtii* adult trees by a triangle (▲).

seedlings and *L. principis-rupprechtii* adult trees (Fig. 4–IV). In sample plot V, there was a negative correlation at a scale of 10–11 m, and there was no correlation at scales of 0–10 m and 11–20 m (Fig. 4–V). In sample plot VI, there was a positive correlation between *L. principis-rupprechtii* seedlings and *L. principis-rupprechtii* adult trees at scales of 2.5–4 m and 8–9 m (Fig. 4–VI), and there was no correlation at other scales.

## Effect of understory herb diversity and canopy density on *L. principis-rupprechtii* seedlings

In these sampling plots, six environment variables (Table 2) could explain 90.2% (PC1: 65.1%; PC2: 25.1%) of the changes found in the number of *L. principis-rupprechtii* seedlings. The number of *L. principis-rupprechtii* seedlings was negatively correlated with the Pielou, Shannon-Wiener, and Simpson indices, and the thickness of the semi-decomposed and undecomposed layers (Fig. 5) was weakly correlated with canopy density. Overall, the understory herb diversity and the thickness of the litter layer significantly affected the number of regenerated *L. principis-rupprechtii* seedlings.

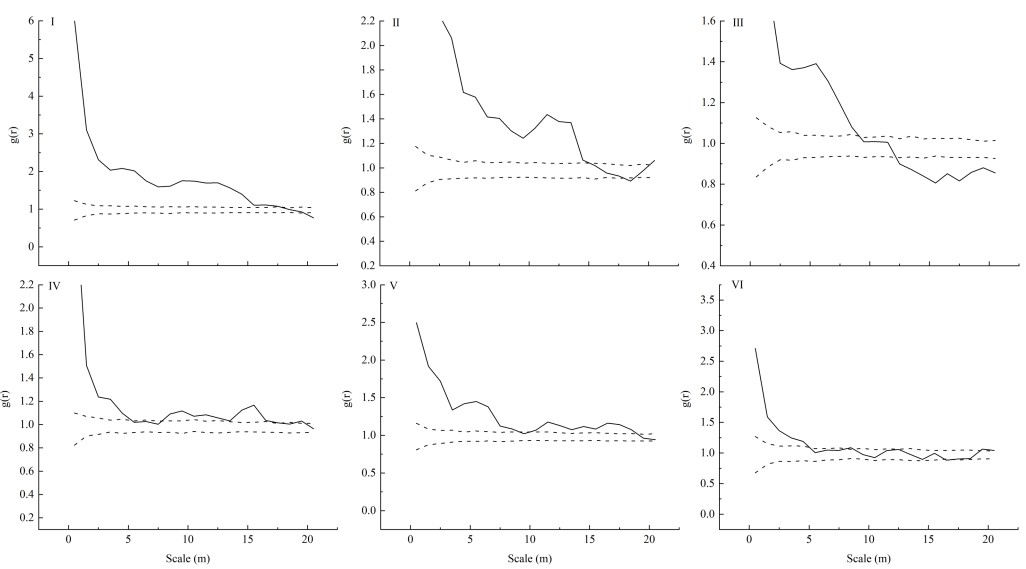

**Figure 3** **Spatial distribution pattern of different plots (I-VI) of *L. principis-rupprechtii* regenerated seedlings in the Guandi Mountains.** r is the radius of the sampling circle with the target tree as the centroid; the upper dotted line represents the upper envelope trace; the lower grey solid line represents the lower envelope trace; the blank middle part represents the 99% confidence interval.

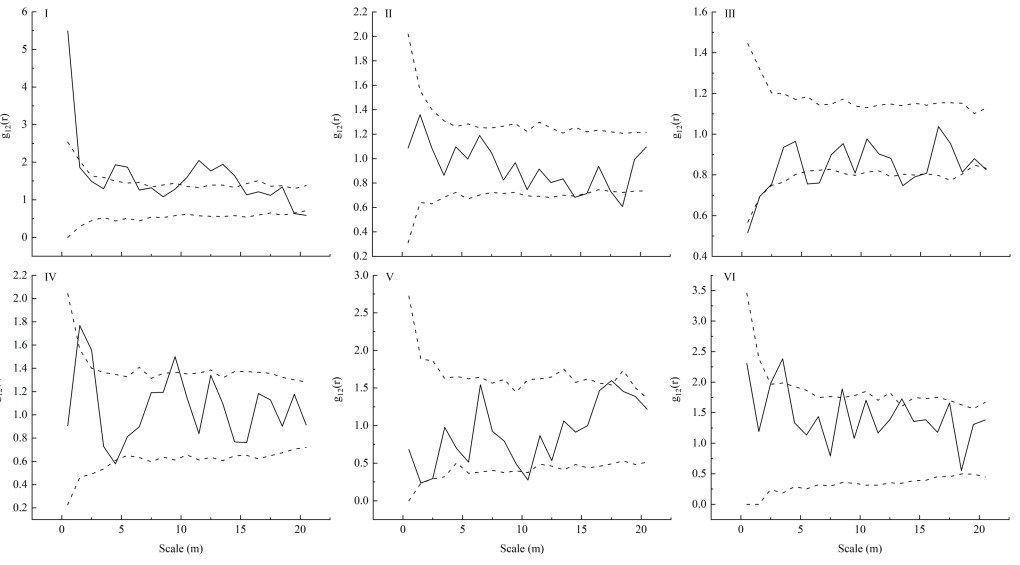

**Figure 4** **Bivariate analysis using null models (independence) for the competition (I-VI) between a pair of *L. principis-rupprechtii* seedlings and *L. principis-rupprechtii* adult trees.** *L. principis-rupprechtii* seedlings formed pattern 1 and random shifts of *L. principis-rupprechtii* adult trees formed pattern 2 across the study region are shown using a torus. Black solid lines represent the function values of the pair-correlation function (g12(r)), upper dotted lines represent the upper envelope trace, lower dotted lines represent the lower envelope trace, and the middle part represents the 99% confidence interval.

**Table 2  The basic information of herb diversity for understory and canopy density in the six sampling plots.**

| Sample plot | *L. principis-rupprechtii* seedlings (number) | The thickness of undecomposed layer (cm) | The thickness of semi-decomposition layer (cm) | Shannon-Wiener index | Pielou index | Simpson index | Canopy density |
|---|---|---|---|---|---|---|---|
| I | 240 | 1.8 | 3.0 | 2.2 | 0.97 | 0.79 | 0.3 |
| II | 327 | 1.5 | 2.3 | 2.4 | 0.98 | 0.80 | 0.2 |
| III | 429 | 1.3 | 1.5 | 2.0 | 0.92 | 0.78 | 0.1 |
| IV | 449 | 1.2 | 1.3 | 2.0 | 0.93 | 0.72 | 0.1 |
| V | 373 | 1.3 | 1.5 | 2.8 | 0.99 | 0.85 | 0.5 |
| VI | 220 | 1.7 | 3.2 | 2.8 | 0.98 | 0.96 | 0.2 |

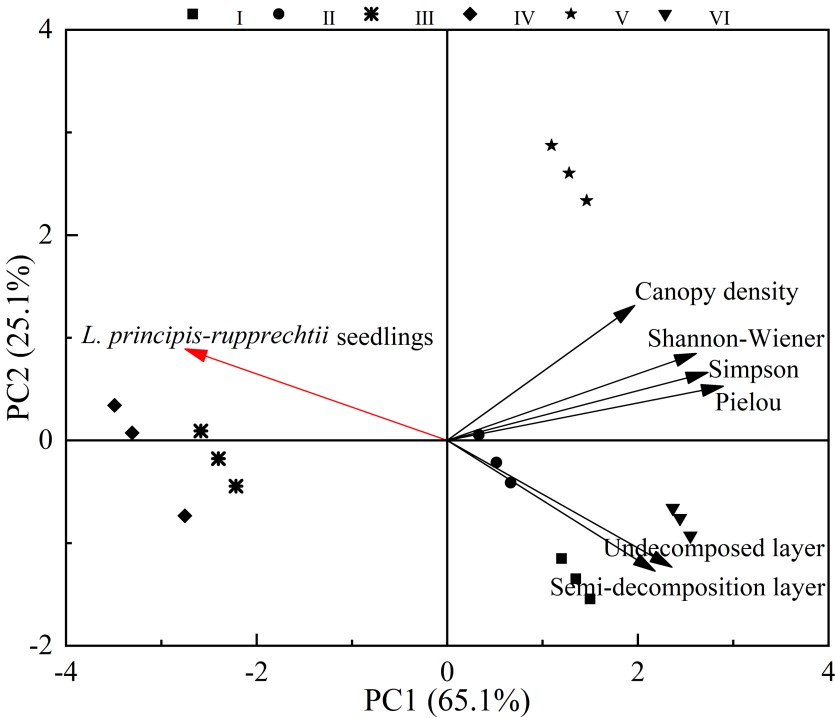

**Figure 5  Principal component analysis of factors affecting the number of *L. principis-rupprechtii* seedlings in different sampling plots.**

## DISCUSSION

### The effect of *L. principis-rupprechtii* seedlings in different sampling plots

Previous studies have shown that the seedling stage is the most important and sensitive stage during forest regeneration (*Deborah & David, 1984*; *Kaoru et al., 1989*; *Wright et al., 2005*), particularly in forests with limited resources and space (*Frost & Rydin, 2000*). We found that the number of *L. principis-rupprechtii* seedlings was mainly determined by reproductive characteristics, as larger populations could produce more high-quality seeds. Natural germinated seedlings are formed when seeds are distributed under a large forest

gap with favorable resources (Fig. 2), and a large number of seedlings are distributed in the community (*Harms et al., 2000*). The demands of *L. principis-rupprechtii* seedlings for light, soil nutrients, and water increase with their growth. *L. principis-rupprechtii* is a typical heliophilous species and their seedlings only grow successfully when occupying a large forest gap. Limited environmental resources and space can increase self-thinning in gradually-matured seedlings (*Condit et al., 2000*). Therefore, to achieve stable population growth, the living environment of *L. principis-rupprechtii* seedlings must be improved.

Species spatial distribution patterns change with scale (*Condit et al., 2000*; *Peter & Chesson, 2000*). The distribution pattern of a species at a small scale is mainly determined by seedling replenishment, while the distribution pattern at a large scale is mainly determined by habitat heterogeneity (*Bai et al., 2012*; *Shen et al., 2013*). In this study, *L. principis-rupprechtii* seedlings across all sample plots showed aggregated distribution at a scale of 0–5 m, and the distribution pattern changed from aggregated distribution to random or regular distribution as the scale increased, which is consistent with the findings of *Zhang & Meng (2007)*. The difference in distribution patterns at the large and small scales was caused by the aggregated distribution at a small scale, which was beneficial for *L. principis-rupprechtii* seedlings to shelter each other, resist the adverse environment, and improve the survival rate. The random or regular distribution at a large scale allowed the *L. principis-rupprechtii* seedlings to avoid intraspecific competition (*e.g.*, for light, water, or other resources) caused by habitat heterogeneity (*Jian et al., 2007*). During the field investigation, we also found that *L. principis-rupprechtii* seedlings were mostly distributed under the forest gap, and there were more seedlings above the litter layer, especially in the thinner thickness of the litter layer. On the other hand, the decreasing number of trees created good conditions for the mass reproduction and growth of young individuals. Additionally, when the root system extended to a certain degree, the self-thinning effect changed its distribution pattern from an aggregate distribution to a random or regular distribution, and *L. principis-rupprechtii* adult trees were mainly in random or regular distributions, which was consistent with the findings of *Jian et al. (2007)* and *Nathan (2006)*.

## Spatial correlation analysis of *L. principis-rupprechtii* seedlings and adult trees

The spatial correlation of individuals of different species within a population shows the spatial distribution and functional relationships among individuals at a particular time (*Shields et al., 2014*). This relationship is the result of the long-term interactions between trees and seedlings, and between the population and environment, and is the measure by which the current situation and population development trend can be predicted. A positive spatial correlation reflects the mutually beneficial ecological relationship within the population while a negative spatial correlation confirms a mutually exclusive ecological relationship within the population (*Zhang et al., 2012*). Previous studies have shown that the spatial distribution characteristics of adult trees affect the spatial correlation of tree growth (*Getzin et al., 2008a*; *Walder & Walder, 2008*). When trees are aggregated, growth reflects an obvious negative correlation, and negative correlation intensity increases with aggregation intensity. *Law et al. (2010)* found that the spatial correlation pattern of tree

growth and the spatial distribution pattern of trees are independent to a certain extent. In this study, we observed no or a weak negative correlation between adult trees and seedlings with the increase of scale (except at a small scale), which may be because adult trees have a lower demand for soil water, light, and other resources and do not compete with seedlings (*Caquet et al., 2010*; *Chen & Cao, 2014*). As a result, there is a large distribution density of individual seedlings around adult trees. The spatial correlations between *L. principis-rupprechtii* seedlings and adult trees in plots V and VI, I and IV, and II and III were similar (Fig. 4), and illustrated the effect of trees on seedlings. We also found that there were few seedlings around trees in some sampling plots (Fig. 2), which was probably caused by limited seed diffusion (*Seidler & Plotkin, 2006*). Because *L. principis-rupprechtii* seeds have seed wings and rely on the wind to spread, most seeds are scattered at a small scale to form seedlings. Additionally, there is less wind in plots at high elevations, which causes scattered distribution of *L. principis-rupprechtii* seeds in these areas. Additionally, the effect of *L. principis-rupprechtii* adult trees on seedlings gradually decreased at a large scale, which shows that the competition between trees and seedlings tends to be gentle and seeds are unable to disperse over a long range. On the other hand, however, this shows the joint action of the bio-ecological characteristics of the population. The findings in this paper are consistent with those of *Zhang et al. (2012)* and *Wang et al. (2017)*.

We also found that individuals with the same tree age were in aggregated distribution. This indicates positive spatial correlation, interdependency, and consistency in the selection of habitats (*Andersen, 1992*) because these individuals are at similar development stages and shelter each other, improving the survival rate. The non-correlation between *L. principis-rupprechtii* adult trees and seedlings at a large scale indicates that the individuals do not interfere with each other in the early development stage. With an increase in scale, the correlation between *L. principis-rupprechtii* adult trees and seedlings became uncorrelated or negatively correlated, indicating that the increasing number of seedlings may lead to growth retardation or death. This explains why there are only a small number of seedlings around *L. principis-rupprechtii* adult trees in some geographical locations (*Getzin et al., 2008b*), which validates our hypothesis (1). Additionally, a few *L. principis-rupprechtii* adult trees produced a better habitat for seedlings, which allowed them to make full use of limited resources in order to grow. Overall, the distance between *L. principis-rupprechtii* seedlings and adult trees affected the natural regeneration of seedlings in the Guandi Mountains. The distance (1.5–4 m) between seedlings and adult trees can be applied to promote the natural regeneration of seedlings. Therefore, the effectiveness and practicality of seedling transplantation using this distance between seedlings and adult trees should be considered in further research.

## The effect of distribution between *L. principis-rupprechtii* adult trees and seedlings

Habitat heterogeneity is considered the most significant factor that leads to seedling aggregation (*Getzin et al., 2008b*). Herb diversity potentially impacts species composition and seedling spatial patterns, and significantly reduces community productivity. We found that herb diversity negatively affected the number of *L. principis-rupprechtii* seedlings,

which decreased the number of *L. principis-rupprechtii* seedlings (Fig. 5 and Table 2), validating our hypothesis (2). Herbs use most of their biomass to produce large and long leaves, increasing the thickness of the litter layer, while reducing the available light, inhibiting the growth of co-occurring species, and impeding the regeneration of seedlings.

An increasing number of herbs increases the competition for soil nutrients among seedlings, affecting species composition. Shading and mechanical hindering (*e.g.*, in twine production) impede seed germination and seedling regeneration and can lead to seedling death. With an increase in herbaceous coverage, the number of seedlings greatly decreased, indicating that herbaceous coverage is the main factor affecting the regenerate degrees of seedlings. The scale of 1.5–4 m was consistent with the range of canopy cover in adult trees. Insufficient light can impede the growth of herbs. A shady environment is optimal for the regeneration of seedlings at an early stage, which further verifies our previous conclusion. Decreased thickness of the semi-decomposed and undecomposed layers lead to a large number of nutrients being decomposed and transferred to the soil to promote the growth of seedlings. According to the results of this study, understory litter is formed at the early stage of regeneration, and the semi-decomposed and undecomposed layers are relatively thin at the scale of 1.5–4 m (*Malik et al., 2020*). The semi-decomposed and undecomposed litter layers showed a negative effect on natural regeneration across the six sampling plots. Seeds germinated into seedlings in the litter layer, but the seed radicles in the thicker litter layer could not reach the soil and obtain sufficient nutrients for growth, leading to the death of many naturally regenerated seedlings in the early development stage (*Nakagawa et al., 2001*; *Nakagawa, Kurahashi & Hogetsu, 2003*). The litter layer was thick around adult trees at a scale of 1 m. However, *L. principis-rupprechtii*'s thick litter layer had an auto-toxic effect on seed germination and seedling growth (*Liira, Sepp & Kohv, 2011*), even though the thick semi-decomposed and undecomposed layers can preserve heat, water, and other resources. Generally, the litter thickness in these plots created an ideal stand for regenerated *L. principis-rupprechtii* seedlings. Thinner and thicker litter layers should be managed in forests. However, the correlation between canopy density and *L. principis-rupprechtii* seedlings was weak, which was consistent with the findings of *Wenjun & Xi* (*2020*) and verifed our hypothesis (3).

## CONCLUSION

This study provided a better understanding of natural regenerative success, and revealed the interactions between populations, environmental factors, and the potential causes affecting regeneration. The univariate distribution type of regenerated seedlings was aggerated at a scale of 0–5 m. The distance between seedlings and adult trees was one of the main drivers for forest succession with a high regeneration ratio. Spatial bivariate analysis showed a positive correlation between seedlings and adult trees, ranging from 1.5 to 4 m. Ideal artificial interference should therefore be considered when determining forest management practices and adjusting the natural regeneration process. In northern China's forest restoration, seedlings should be transplanted at a proper distance from the adult trees in order to stimulate the natural regeneration of *L. principis-rupprechtii*. Herb diversity

and litter thickness pose significant negative influences on the number of regenerated seedlings. Herbs, shrubs, and unhealthy and dead trees should be removed appropriately. Extra-thick litter layers should be promptly removed to ensure a proper litter thickness. These intervention strategies can reduce the competition between enrichment species and regenerated seedlings, allowing for natural regeneration and long-term sustainable development.

## ACKNOWLEDGEMENTS

We thank anonymous reviewers for the useful comments.

### Funding

This work was supported by the National Natural Science Foundation of China (No. 31971644,31901365, and 31500523), the Innovation Project of Graduate Education in Shanxi Province (No. 2020BY048), the Technological Innovation Project of Colleges and Universities in Shanxi Province (No. 2019l0394), the Shanxi Provincial Outstanding Doctoral Program for Incentive Funds for Scientific Research Projects (No. SXYBKY2018032), and the Fund for Introduced Talents for Shanxi Agricultural University (No. 2018yj09 and 2014yj19). The funders had no role in study design, data collection and analysis, decision to publish, or preparation of the manuscript.

### Grant Disclosures

The following grant information was disclosed by the authors:
The National Natural Science Foundation of China: 31971644, 31901365, 31500523.
Innovation Project of Graduate Education in Shanxi Province: 2020BY048.
Technological Innovation Project of Colleges and Universities in Shanxi Province: 2019l0394.
Shanxi Provincial Outstanding Doctoral Program for Incentive Funds for Scientific Research Projects: SXYBKY2018032.
Shanxi Agricultural University: 2018yj09, 2014yj19.

### Competing Interests

The authors declare there are no competing interests.

### Author Contributions

- Weiwen Zhao conceived and designed the experiments, performed the experiments, analyzed the data, prepared figures and/or tables, authored or reviewed drafts of the paper, and approved the final draft.
- Wenjun Liang conceived and designed the experiments, performed the experiments, analyzed the data, authored or reviewed drafts of the paper, and approved the final draft.
- Youzhi Han conceived and designed the experiments, performed the experiments, authored or reviewed drafts of the paper, and approved the final draft.

- Xi Wei conceived and designed the experiments, performed the experiments, authored or reviewed drafts of the paper, and approved the final draft.

## Data Availability

The raw measurements are available in the Supplemental File.

## Supplemental Information

Supplemental information for this article can be found online at http://dx.doi.org/10.7717/peerj.12327#supplemental-information.

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
