# Peer review of "Characteristics and factors influencing the natural regeneration of Larix principis-rupprechtii seedlings in northern China"

_PeerJ, doi:10.7717/peerj.12327_

## Round 0.1 · original submission · Major Revisions

Dear Dr. Zhao,

After two independent reviews, I believe your manuscript may be published in PeerJ after you take care of the issues raised by both reviewers. Please take special care with the issues raised by reviewer #2.

As soon as you can resubmit, please do not forget to prepare a rebuttal letter informing the reviewers about the main changes you performed to the text.

Sincerely, Daniel Silva.

Reviewer 1 ·

Basic reporting

Clear English used throught, enought literature for background, Good structrue and figures and tables

Experimental design

clear research question, and correct research methods

Validity of the findings

no comments

Additional comments

The natural regeneration of Larix principis-rupprechtii is still a problem in northern China. Understanding natural regeneration is important for forest management, thus your study could be interesting for the associated reader. The authors used the point pattern analysis to illustrate the relationship of seedlings and adult trees of L. principis-rupprechtii and then, analyzed the potential factors that affect regeneration. This seems to be of significance to the instruction for sustainable management of natural regeneration in mountains of northern China. However, there are several minor problems need to be explained and modified.
1. What is the historic vegetation cover of the GDM and why is it a target for afforestation efforts?
2. Why was L. principis-repprechtii chosen for afforestation project in this region?
3. The diversity indices you mentioned in the results could not be found in methods. For example, you mentioned the undergrowth vegetation and litter, but it was unclear what methods did you use. I suggest explicitly starting how you sampled the herbaceous community, diversity indices and litter indices you used to assess.
4. In the objectives you should mention your target species. The first two hypothesis/questions have been answered for other tree species, as you can see from the literature. The third objective can be removed or changed to "identify other environmental factors affecting natural regeneration..."
5. In the Method section you have to clarify whether you study natural forests or plantations. Were these forests recently thinned. Are these forests managed and/or burnt?
6. Please check the labelling of Fig. 2. You might have switched trees and seedlings. I thought seedlings would be more abundant.
7. I’m concerned with the location of litter and herb samples in plots. Are samples taken next to each seedling and adult trees?
a. line 123. Authors mentioned in M&M that the study was carried out in the summer of 2020. Please indicate the date of the start and the end of the field samplings.
b. line 140. How many litters and herbs did you sample?
8. The authors indicated litter and herb effected the regenerated seedlings; it is unclear that how to influence regeneration. Authors stated only some presumption in term of the seed germination but these were not confirmed by examinations or scientific references.
9. In addition to some descriptive sampling methods, you use the word “significantly” in line 229, please add p-values or effect sizes in Fig. 5.
10. Dong et al (https://doi.org/10.1007/s40333-018-0004-3) have shown that the function g(r) has more disadvantages compared with the function O(r). it is suggested that the author should consider the function with better performance here.
11. Generally speaking, the relatively distance between seedling and adult tree effect regeneration, soil nutrient and light are limited; from Fig. 2, some adult trees are distribution in plots, however, the authors use the g12(r) illustrate the relationship between seedling and adult trees, it is unclear that which one does it represent g1 and g2; in Discussion, the authors mention the scale of 1.5-4 m between seedling and adults can promote natural regeneration. Whether the scale is correct.
12. Keywords should not be the repetitions of the title words, please find words which are not in the title, this way search engines of the web will find your manuscript with higher probability.
13. Summarizing the constituents of the common vegetation would be favorable in a tabular form.
14. Please clarify whether the floristic composition survey was conducted by the authors or not? If not, please indicate the source of the data.

Reviewer 2 ·

Basic reporting

The manuscript named “Characteristics and influencing factors of the natural regeneration of Larix principis-rupprechtii seedlings in northern China” submitted by Zhao et al. conducted the interesting work regarding to the spatial patterns analysis of Larix principis-rupprechtii seedlings natural regeneration in North China forest ecosystems. Its aim are novel and important to reveal the factors which are mainly responsible for the natural regeneration of Larix- the important afforestation tree species in North China for future study. The results sound very interesting and novel. However, I still have severely major concerns with the experiment design itself and some severe flaw also for the writing importance particularly for its introduction and discussion sections if they can induce to the robust and solid conclusion from the current experimental design.

Experimental design

The current descriptions of experimental design, in my opinion, are not efficient to introduce a clear results. e.g. What is the exactly size of selected random plots? Why you selected in the beginning for 40 plots? from the Fig. 1 plot map information, I doubt why selected the plots very close to the uncovered land-system (No. 2 and 3 plots)? which could have great influence on the results of investigation. The M& M section lack of details description for root and litter analysis parts.

Validity of the findings

Conclusion is general ok but is too brief to related to current results.

Additional comments

The manuscript named “Characteristics and influencing factors of the natural regeneration of Larix principis-rupprechtii seedlings in northern China” submitted by Zhao et al. conducted the interesting work regarding to the spatial patterns analysis of Larix principis-rupprechtii seedlings natural regeneration in North China forest ecosystems. Its aim are novel and important to reveal the factors which are mainly responsible for the natural regeneration of Larix- the important afforestation tree species in North China for future study. The results sound very interesting and novel. However, I still have severely major concerns with the experiment design itself and some severe flaw also for the writing importance particularly for its introduction and discussion sections if they can induce to the robust and solid conclusion from the current experimental design.
e.g. the introduction section, the author only focused to introduce the importance for Larix in GDM area, which was too regional to be suitable to expand the results to more interesting audiences.

Annotated reviews are not available for download in order to protect the identity of reviewers who chose to remain anonymous.

---

## Round 0.2 · Major Revisions

Dear Dr. Zhao,

Considering the review by one previous reviewer of your study, who decided on its acceptance, I made a quick reading of it, and I also find the current version satisfactory.

Nonetheless, after a in-house review, it was detected that some text improvements are needed. For instance, there are a few sentences that appear incomplete and should be rewritten (lines 39, 55, 57; maybe others). As well, there are some examples where the incorrect tense or word usage is used (lines 62, 66, 71, 74, 109, 123, 192; maybe others). There is no reference for the software used (line 183). The formatting of Table 1 requires adjustment. The manuscript should be reviewed again for clarity (perhaps with assistance)

Therefore, although academically the reviewers considered the text is acceptable, Further text improvements are needed. Therefore, I am returning the text to you, for a new review of the written English in your study.

Sincerely,
Daniel Silva

Reviewer 1 ·

Basic reporting

excellent

Experimental design

correct

Validity of the findings

Conclusions are well stated

Additional comments

I am generally satisfied with this version of the manuscript revised by the authors. The quality of this work has been substantially improved. I suggest accepting this work for publication in PeerJ.

---

## Round 0.3 · Minor Revisions

Dear Zhao et al.,

After another reading of your text, I detected that significant written English issues persisted. Therefore, I am returning the text to you, further review of written English in your text. For the text to be formally accepted for publication in PeerJ, I gently require from you proofs that a fluent-English-speaking colleague or a professional editing service has polished the language.

I am not a native speaker, and I am aware of the issues non-native speakers pass through when they do science that needs to be published in languages other than our own. Still, I am requesting this measure from you to maintain the quality of the studies published in PeerJ.

Sincerely,
Daniel Silva, Ph.D.

---

## Round 0.4 · accepted · Accept

Dear Dr. Zhao,

Thank you for your persistence! I now consider your manuscript as formally accepted for publication in PeerJ!

Sincerely, Daniel Silva.